# Proposal: Transformer Models for Predicting Material Properties

**Kaiwei Zhang**
School of Materials Science and Engineering

**Juncheng Yu**
School of Aerospace Engineering

**Haonan Li**
Department of Power and Energy Engineering

## 1   Background

In recent years, artificial neural networks have emerged as a novel approach to acquire the properties of materials. Models like SchNet [1], CGCNN [2], MEGNet [3], ALIGNN [4], etc. have reached high accuracies on various properties such as formation energy, moduli, band gap, etc. They have also been used to discover new materials [5], guide material synthesis [6], uncover structure-property relationships [7], and are also proven useful in predicting many specific properties in refined areas [8, 9].

Currently, leading models are mostly GNNs and such architecture has long dominated the field. A node in the graph represents an atom, whereas an edge represents a bond. A problem that GNNs have is how to assign the contribution of different sites and atoms to the predicted value. For example, when it comes to perovskite, a material with the formula ABX3 as shown in Figure 1(a), it is difficult to determine whether the A site is more important to the bandgap, if the B site influences energy, or whether the X element matters. This difficulty arises because global pooling ignores the diversity of nodes in the graph.

Another problem of current models is that they are not adept at capturing global features of crystal structures. For a graph with N nodes, it could take $O(N)$ layers for a node to receive the features of the farthest node if the nodes are sparsely connected, which is the case if materials are viewed as crystal graphs. Nevertheless, these models are not deep enough for information to travel across the whole graph, as shown in Figure 1(b). As a result, current models are not exceptional at properties related to the global features of a material. A typical example of a material's global feature is its symmetry (shown in Figure 1(c)), which is strongly related to properties such as piezoelectricity and dielectricity. For a material to be piezoelectric, it must lack inversion symmetry. The dielectric constant tends to be higher when materials possess low symmetry. A good prediction of piezoelectric and dielectric constants must start with assessing the global symmetry. As current models lack a global viewpoint, they perform unsatisfactorily on such tasks. For instance, ALIGNN [4] predicts formation and total energy much better than piezoelectric and dielectric constants. Energy prediction requires less global information, as one can estimate the total energy by adding up the energies of each atom [10].

Transformers [11] may be applied to the prediction of crystal properties. We can simply view the atoms in a material's lattice as a set of input tokens, and the coordinates can be treated by using methods similar to positional encoding. By using the attention mechanism, transformers are well-suited for tackling the aforementioned problems. The attention matrix could be utilized to assess the individual contribution and atomic interactions. On the other hand, the maximum path length of attention mechanism is $O(1)$ [11], making it ideal for grasping global features. It should also be noted that transformers are flexible to the length of the input, which is consistent with the nature of materials considering that the number of atoms inside a lattice could vary substantially.

Preprint. Under review.

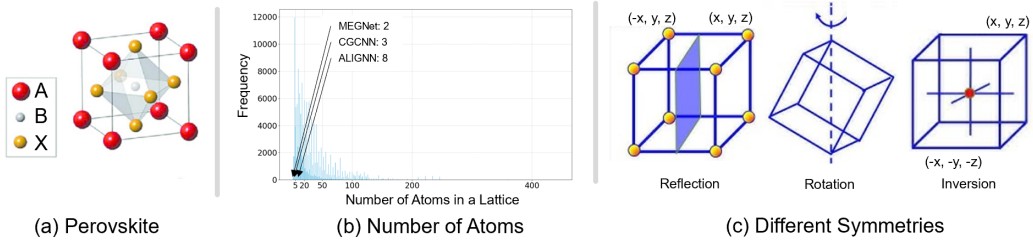

(a) Perovskite          (b) Number of Atoms          (c) Different Symmetries

Figure 1: Illustrations

## 2 Related work

**GNN models** Various GNN models have been developed to predict the properties of materials. CGCNN [2] uses graph convolutional layers and is interpretable in a few cases by observing local chemical environments. The MEGNet [3] uses the algorithm of graph networks [12] of sequentially updating the bonds, atoms, and global attributes of a material, resulting in improved behavior. ALIGNN [4] further takes into account the impact of angles between bonds. It involves two parallel graphs with linking message pathways. Models such as BonDNet [13] and DimeNet [14] also belong to the GNN family, yet they are developed for molecules and do not work well on crystals.

**Attention mechanism** There have been attempts to apply attention mechanisms. GATGNN [15] uses several augmented graph attention layers (AGAT) to learn the local relationship between adjacent atoms and a global attention layer is applied at last. CTGNN [16] uses transformer aside from GNN, yet its prediction is limited to energies. It would be reluctant to recognize these models as fully effective solutions for the problems of contribution assignment and global information extraction.

## 3 Proposed method

**Model training** Our research consists of four parts. To start with, we will train a model for predicting material properties. In terms of model structure, we are hoping for a transformer model. Methods like structural encoding [17] could help transformer models adapt to graphic tasks. As for training data, Materials Project [18] is a dataset that holds calculated properties of over 150000 materials. OQMD [19], JARVIS-DFT [20] and Google's GNoME [21] could also enrich the potential dataset. Existing benchmarks [22] will be used to evaluate our model's capabilities in predicting formation energies, bandgaps, moduli, etc.

**Attention analysis** Secondly, we expect to assign contributions of each atom by analyzing the attention weights. Following the design of BERT [23], an additional classifier token `[cls]` could be applied to track the individual influences on the final output. If the results are ideal, they could be used to observe the contribution of different sites and elements inside a crystal. Besides, attention could also be used to study the interactions between atoms. We hope that the model could intrinsically generate higher attention weights between atoms that have smaller distances or stronger chemical bonds. To achieve a better interpretation of the learned attention, techniques including attention rollout, attention flow [24], and gradient self-attention [25] maps could be utilized.

**Global view demonstration** Thirdly, we wish to demonstrate the transformer's advantage of capturing global information. The model's ability to handle symmetry information could be illustrated by comparing the distribution of encoded vectors with point group categories. We also expect our model to reach better accuracy at predicting properties related to global information, such as piezoelectric dielectric constant.

**Further discussion** Lastly, if the previous sections are smoothly accomplished, we hope to further investigate the potential of our model. We could compare it with fine-tuned LLMs [26, 27] in sorting tasks, and investigate the element embeddings [10] for chemical instincts. We could also use our model to discover novel materials within a ternary or quaternary system [5].

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
