# OpenReview forum: "【Proposal】Transformer Models for Predicting Material Properties"
_tsinghua.edu.cn/THU/2024/Fall/AML — THU 2024 Fall AML Submission_

### Official Review · ~Ethan_Wei_Yuxin1 · 2024-11-08
**Well done!**

**Rating:** 8
**Confidence:** 3

**Review:**

Your proposal presents a compelling approach to enhancing material property prediction by addressing key limitations in current graph neural network (GNN) models. You effectively highlight challenges such as the difficulty of capturing global features and assigning individual contributions to atoms within materials. Your proposed transformer model, with its attention mechanisms, offers a thoughtful solution to these challenges, aiming to improve accuracy in predicting properties that depend on global structural features, like piezoelectric and dielectric constants. The step-by-step methodology— including attention analysis, model training, and the demonstration of global information capture—shows a well-planned strategy that leverages transformers' strengths for this domain. This is a promising direction, and I look forward to seeing how your work advances the field of material property prediction.

---

### Official Review · ~Zhen_Leng_Thai1 · 2024-11-08
**Well-Structured Proposal on Transformer-Based Material Properties Prediction**

**Rating:** 9
**Confidence:** 4

**Review:**

This paper proposes a transformer model with an attention mechanism to predict material properties, addressing accuracy issues in GNN models for this task. The problem definition is clear, and the proposed method is well-structured. However, the related works section overlooks some relevant transformer-based studies, such as Materials Informatics Transformer: A Language Model for Interpretable Materials Properties Prediction.

---

### Official Review · ~Un_Lok_Chen1 · 2024-11-08
**A Transformer-based Model that Better Captures Global Features in Predicting Material Properties**

**Rating:** 9
**Confidence:** 4

**Review:**

Summary:

This proposal explores the potential of Transformer models in predicting material properties. The authors suggest that compared with GNNs, Transformer models may excel in capturing global features, dealing with variable numbers of atoms in different molecules, and attributing global property contribution to individual atoms. They attempt to adopt some attention analysis techniques to better understand how the Transformer models may be useful.

Pros:
1. In the Background section, the authors present the problem context with abundant examples accompanied by clear explanations, thus making the reasoning about using Transformer to better capture global features of the molecule easy to follow.
2. The proposal includes a detailed description of potential analysis (e.g. attention weight visualization) that can be conducted, which seems reasonable enough; the step-by-step implementation plan is also well-thought-out.

Cons:

A. Major issues
1) Are there any existing works that also adopt the Transformer model approach to predict the material properties or in similar scientific tasks?

B. Minor issues

1) The caption for Figure 1 can be more specific.

2) Please include in-text citations for the figures if they are adopted from other works.

3) The “structural encoding” method mentioned in the Model training part of the Proposed Method section may require further elaboration if it is a crucial component of the model.

4) Would comparison experiments be conducted with the proposed Transformer model (e.g. the GNN architecture w/ attention mechanism as mentioned in the Related Work section)?

---

### Official Review · ~André_Moreira_Leal_Leonor1 · 2024-11-09
**Good job**

**Rating:** 9
**Confidence:** 4

**Review:**

The proposal effectively argues for using transformer models to improve material property prediction, addressing GNN limitations in capturing atomic contributions and global features. It’s well-researched, with strong points on transformer benefits for handling global structures and input flexibility.

Strengths:

Clear articulation of GNN challenges and transformer advantages.
Solid related work and methodology.

Areas to Improve:

Explain how global symmetry will be encoded.
Clarify performance metrics and address transformers' computational demands.
Suggestions:

Add a specific example, like predicting bandgap, to illustrate the approach.

---

### Official Review · ~Chan_Thong_Fong1 · 2024-11-10
**Innovative idea to implement transformer models in predicting material properties**

**Rating:** 8
**Confidence:** 4

**Review:**

The proposed method offers a compelling advancement in material property prediction by leveraging transformers, which have the potential to overcome key limitations of current graph neural network (GNN)-based models. This approach could significantly enhance the interpretability and accuracy of predictions compared to GNN models, which struggle with global feature extraction and atom-specific contributions. Additionally, the proposed framework’s flexibility in handling input size and its potential for discovering new materials make it an exciting avenue for future material science research. However, the method’s success will depend on the effective integration of structural encoding and attention analysis, which will need to be rigorously validated through comprehensive experimentation on large, diverse datasets.

---

### Official Review · ~Tianxing_Yang1 · 2024-11-11
**Evaluating the Use of Transformer Models for Predicting Material Properties**

**Rating:** 9
**Confidence:** 4

**Review:**

This proposal suggests using transformer models to predict the properties of materials.

Pros:
- Utilizing machine learning methods for material property prediction represents a novel research paradigm with significant potential.
- The proposed methodology section outlines a fairly detailed, step-by-step research plan.

Cons:
- The application of attention mechanisms to graph-related tasks has been widely explored in previous studies [1][2][3]. If the author intends to introduce innovation in the model architecture, it would be beneficial to discuss how the proposed approach differs from existing methods in the final paper.
- Addressing issues such as the assessment of global symmetry in materials could be approached by treating it as a downstream application of graph matching tasks.




References:

[1] Petar Veličković et al., Graph Attention Networks

[2] Seongjun Yun et al., Graph Transformer Networks

[3] Chengxuan Ying et al., Do Transformers Really Perform Bad for Graph Representation?

---

### Official Review · ~Liu_Yiyang1 · 2024-11-11
**Innovative and well pitched project**

**Rating:** 9
**Confidence:** 3

**Review:**

This paper proposes the use of transformer models in predicting material properties, which is an area currently dominated by GNN models. It presents a compelling argument towards the potential benefits of using transformer models, diving into great detail regarding the shortcomings of GNN models and how these shortcomings can be mitigated with the use of transformer models. The proposed methodology  also included concrete details on areas like datasets, design methodologies and potential optimization strategies. One potential area of concern is that accurately capturing 3D spatial relations and symmetry into a 1D positional encoding scheme accurately might be tricky and need to be extensively validated in the future. Otherwise, very well pitched and thoroughly research proposal!

---

### Official Review · ~Chaoqun_Yang2 · 2024-11-12
**Good questions and appropriate methods**

**Rating:** 10
**Confidence:** 4

**Review:**

**Summary:**
The proposal presents an innovative approach to predicting material properties using transformer models and exploring the explainability of the model according to attention weights.

**Highlights:**
1. **Clear Problem Introduction:** The proposal clearly articulates the limitations of current GNN models in materials science, specifically their challenges in assigning atomic contributions and capturing global features of crystal structures. This sets a strong foundation for the proposed research.
2. **Appropriate Technical Selection:** The choice of transformer models is well-justified, given their success in handling long-range dependencies and global context in other domains. This selection is appropriate for addressing the identified limitations of GNNs in materials property prediction.
3. **Logical Method Design:** The methodology is logically designed, starting from model training with substantial datasets to analyzing attention weights and demonstrating the model's global view capabilities. This step-by-step approach is sound and addresses the research objectives effectively.

**Advice:**
1. **Model Reliability:** A critical aspect to address is the reliability of the transformer model's predictions. The proposal hinges on the premise that transformers can accurately predict material properties. It is essential to validate the model's prediction accuracy rigorously. If the transformer model's accuracy is mediocre, the insights derived from attention weights may be unreliable.

---

### Official Review · ~Kairong_Luo1 · 2024-11-12
**Interdisciplinary idea**

**Rating:** 8
**Confidence:** 2

**Review:**

Strength:
1. quite an interdisciplinary idea;
2. a detailed description about the problem setting and possible approaches;
3. the related work review is very thorough.

Weakness:
1. i am curious about the paradigm in this field. It seems just mapping the material problem into a graphical problem. Can more material information be encoded into the structure or embedding? i do not know whether there is some hint.
2. the advantage in LLM seems not explained in details. A noteworthy problem is that the LLM needs great data, even in fine-tuning. Maybe the volume of dataset could be shown.

---

### Official Review · ~Zihan_Wang7 · 2024-11-12
**meaningful material properties predicting model**

**Rating:** 10
**Confidence:** 3

**Review:**

**Highlights**

* Clearly identifies two key limitations of current GNN models, demonstrating their practical impact through specific cases such as perovskites.

* The attention mechanism naturally matches the needs of materials science, with solutions supported by theoretical foundation and feasibility.

**Advice**

* Suggest supplementing theoretical analysis of how Transformer handles 3D spatial data, and explain how to integrate crystallographic knowledge into the model architecture.